# CentripetalText: An Efficient Text Instance Representation for Scene Text Detection

**Tao Sheng, Jie Chen, Zhouhui Lian**[*]
Wangxuan Institute of Computer Technology
Peking University, Beijing, China
{shengtao, jiechen01, lianzhouhui}@pku.edu.cn

## Abstract

Scene text detection remains a grand challenge due to the variation in text curvatures, orientations, and aspect ratios. One of the hardest problems in this task is how to represent text instances of arbitrary shapes. Although many methods have been proposed to model irregular texts in a flexible manner, most of them lose simplicity and robustness. Their complicated post-processings and the regression under Dirac delta distribution undermine the detection performance and the generalization ability. In this paper, we propose an efficient text instance representation named CentripetalText (CT), which decomposes text instances into the combination of text kernels and centripetal shifts. Specifically, we utilize the centripetal shifts to implement pixel aggregation, guiding the external text pixels to the internal text kernels. The relaxation operation is integrated into the dense regression for centripetal shifts, allowing the correct prediction in a range instead of a specific value. The convenient reconstruction of text contours and the tolerance of prediction errors in our method guarantee the high detection accuracy and the fast inference speed, respectively. Besides, we shrink our text detector into a proposal generation module, namely CentripetalText Proposal Network (CPN), replacing Segmentation Proposal Network (SPN) in Mask TextSpotter v3 and producing more accurate proposals. To validate the effectiveness of our method, we conduct experiments on several commonly used scene text benchmarks, including both curved and multi-oriented text datasets. For the task of scene text detection, our approach achieves superior or competitive performance compared to other existing methods, e.g., F-measure of 86.3% at 40.0 FPS on Total-Text, F-measure of 86.1% at 34.8 FPS on MSRA-TD500, etc. For the task of end-to-end scene text recognition, our method outperforms Mask TextSpotter v3 by 1.1% in F-measure on Total-Text.

## 1 Introduction

In the past decade, scene text detection has attracted increasing interests in the computer vision community, as localizing the region of each text instance in natural images with high accuracy is an essential prerequisite for many practical applications such as blind navigation, scene understanding, and text retrieval. With the rapid development of object detection [29, 19, 7, 18] and segmentation [22, 46, 26, 50], many promising methods [51, 23, 37, 16, 38, 39] have been proposed to solve the problem. However, scene text detection is still a challenging task due to the variety of text curvatures, orientations, and aspect ratios.

How to represent text instances in real imagery is one of the major challenges for scene text detection, and usually there are two strategies to solve the problem arising from this challenge. The first is to treat text instances as a specific kind of object and use rotated rectangles or quadrangles for

---

[*]Corresponding author

35th Conference on Neural Information Processing Systems (NeurIPS 2021).

description. This kind of methods are typically inherited from generic object detection and often utilize manually designed anchors for better regression. Obviously, this solution ignores the geometric traits of irregular texts, which may introduce considerable background noises, and furthermore, it is difficult to formulate appropriate anchors to fit the texts of various shapes. The other strategy is to decompose text instances into several conceptual or physical components, and reconstruct the polygonal contours through a series of indispensable post-processing steps. For example, PAN [38] follows the idea of clustering and aggregates text pixels according to the distances between their embeddings. In TextSnake [23], text instances are represented by text center lines and ordered disks. Consequently, these methods are more flexible and more general than the previous ones in modeling. Nevertheless, most of them suffer from slow inference speed, due to complicated post-processing steps, essentially caused by this kind of tedious multi-component-based representation strategy. For another, their component prediction is modeled as a simple Dirac delta distribution, which strictly requires numerical outputs to reach the exact positions and thus weakens the ability to tolerate mistakes. The wrong component prediction will propagate errors to heuristic post-processing procedures, making the rebuilt text contours inaccurate. Based on the above observations, we can find out that the implementation of a fast and accurate scene text detector heavily depends on a simple but effective text instance representation with a robust post-processing algorithm, which can tolerate ambiguity and uncertainty.

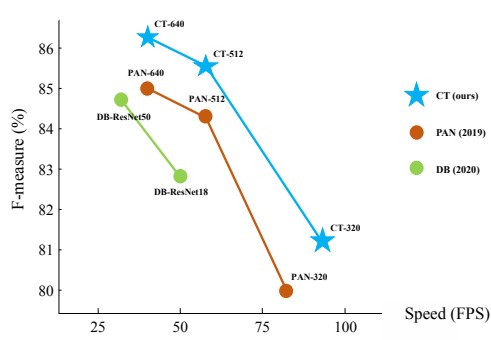

Figure 1: Performance and speed of some top-performing real-time scene text detectors on the Total-Text dataset. Enabled by the proposed CT, an efficient representation of text instances, our method outperforms DB [16] and PAN [38], and achieves the best trade-off between accuracy and speed. More results are shown in Tab. 3.

To overcome these problems, we propose an efficient component-based representation method named CentripetalText (CT) for arbitrary-shaped texts. Enabled by the proposed CT, our scene text detector outperforms other state-of-the-art approaches and achieves the best trade-off between accuracy and speed (as shown in Fig. 1). Specifically, as illustrated in Fig. 4, our method consists of two steps: i) input images are fed to the convolutional neural network to predict the probability maps and centripetal shift maps. ii) pixels are grouped to form the text instances through the heuristics based on text kernels and centripetal shifts. In details, the text kernels are generated from the probability map followed by binarization and connected component search, and the centripetal shifts are predicted at each position of the centripetal shift map. Then each pixel is shifted by the amount of its centripetal shift from its original position in the centripetal shift map to the text kernel pixel or background pixel in the probability map. All pixels that can be shifted into the region of the same text kernel form a text instance. In this manner, we can reconstruct the final text instances fast and easily through marginal matrix operations and several calls to functions of the OpenCV library. Moreover, we develop an enhanced regression loss, namely the Relaxed L1 Loss, mainly for dense centripetal shift regression, which further improves the detection precision. Benefiting from the new loss, our method is robust to the prediction errors of centripetal shifts because the centripetal shifts which can guide pixels to the region of the right text kernel are all regarded as positive. Besides, CT can be fused with CNN-based text detectors or spotters in a plug-and-play manner. We replace SPN in Mask TextSpotter v3 [13] with our CentripetalText Proposal Network (CPN), a proposal generation module based on CT, which produces more accurate proposals and improves the end-to-end text recognition performance further.

To evaluate the effectiveness of the proposed CT and Relaxed L1 Loss, we adopt the design of network architecture in PAN [38] and train a powerful end-to-end scene text detector by replacing its text instance representation and loss function with ours. We conduct extensive experiments on the commonly used scene text benchmarks including Total-Text [2], CTW1500 [47], and MSRA-TD500 [45], demonstrating that our method achieves superior or competitive performance compared to the state of the art, e.g., F-measure of 86.3% at 40.0 FPS on Total-Text, F-measure of 86.1% at 34.8 FPS on MSRA-TD500, etc. For the task of end-to-end text recognition, equipped with CPN, the

F-measure value of Mask TextSpotter v3 can further be boosted to 71.9% and 79.5% without and with lexicon, respectively.

Major contributions of our work can be summarized as follows:

- We propose a novel and efficient text instance representation method named CentripetalText, in which text instances are decomposed into the combination of text kernels and centripetal shifts. The attached post-processing algorithm is also simple and robust, making the generation of text contours fast and accurate.
- To reduce the burden of model training, we develop an enhanced loss function, namely the Relaxed L1 Loss, mainly for dense centripetal shift regression, which further improves the detection performance.
- Equipped with the proposed CT and the Relaxed L1 Loss, our scene text detector achieves superior or competitive results compared to other existing approaches on the curved or oriented text benchmarks, and our end-to-end scene text recognizer surpasses the current state of the art.

## 2 Related work

Text instance representation methods can be roughly classified into two categories: component-free methods and component-based methods.

**Component-free methods** treat every text instance as a complete whole and directly regress the rotated rectangles or quadrangles for describing scene texts without any reconstruction process. These methods are usually inspired by general object detectors such as Faster R-CNN [29] and SSD [19], and often utilize heuristic anchors as prior knowledge. TextBoxes [15] successfully adapted the object detection framework SSD for text detection by modifying the aspect ratios of anchors and the kernel scales of filters. TextBoxes++ [14] and EAST [51] could predict either rotated rectangles or quadrangles for text regions with and without the prior knowledge of anchors, respectively. SPCnet [40] modified Mask R-CNN [7] by adding the semantic segmentation guidance to suppress false positives.

**Component-based methods** prefer to model text instances from local perspectives and decompose instances into components such as characters or text fragments. SegLink [31] decomposed long texts into locally-detectable segments and links, and combined the segments into whole words according to the links to get final detection results. MSR [43] detected scene texts by predicting dense text boundary points. BR [30] further improved MSR by regressing the positions of boundary points from two opposite directions. PSENet [37] gradually expanded the detected areas from small kernels to complete instances via a progressive scale expansion algorithm.

## 3 Methodology

In this section, we first introduce our new representation (i.e., CT) for texts of arbitrary shapes. Then, we elaborate on our method and training details.

### 3.1 Representation

**Overview** An efficient scene text detector must have a well-defined representation for text instances. The traditional description methods inherited from generic object detection (e.g., rotated rectangles or quadrangles) fail to encode the geometric properties of irregular texts. To guarantee the flexibility and generality, we propose a new method named CentripetalText, in which text instances are composed of text kernels and centripetal shifts. As demonstrated in Fig. 2, our CT expresses a text instance as a cluster of the pixels which can be shifted into the region of the same text kernel through centripetal shifts. As pixels are the basic units of digital images, CT has the ability to model different forms of text instances, regardless of their shapes and lengths.

Mathematically, given an input image $I$, the ground-truth annotations are denoted as $\{T_1, T_2, .., T_i, ..\}$, where $T_i$ stands for the $i$th text instances. Each text instance $T_i$ has its corresponding text kernel $K_i$, a shrunk version of the original text region. Since a text kernel is a subset of its text instance, which

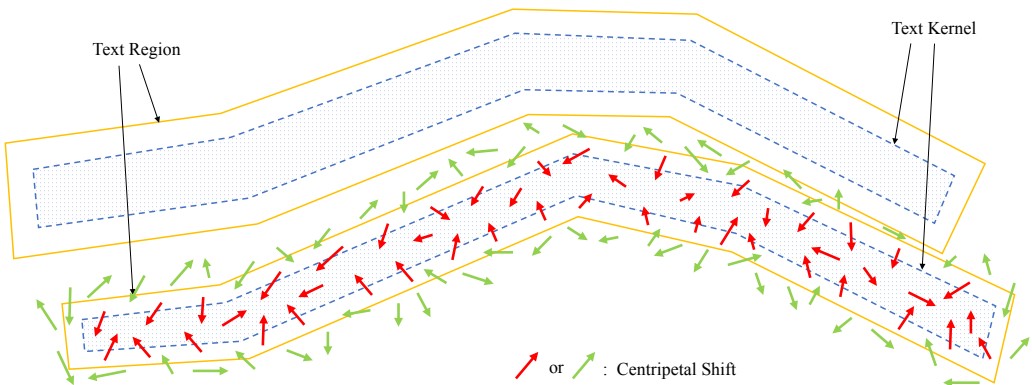

Figure 2: Illustration of the proposed CentripetalText representation. Text regions (in yellow) can be decomposed into the combination of text kernels (in blue) and centripetal shifts (either in red or green). The centripetal shifts represented as green arrows start from background pixels to non-text-kernel pixels, which are useless for the further generation of text contours, while other centripetal shifts in red start from text region (foreground) pixels to text kernel pixels, which contribute to define the shapes. All the pixels that can be shifted into the region of the same text kernel form a text instance. For the sake of better demonstration, we only visualize the centripetal shifts over the bottom text instance.

satisfies $K_i \subseteq T_i$, we treat it as the main basic of the pixel aggregation. Different against the distance in conventional methods, each centripetal shift $s_j$ which appears in a position $p_j$ of the image guides the clustering of text pixels. In this sense, the text instance $T_i$ can be easily represented with the aggregated pixels which can be shifted into the region of a text kernel according to the values of their centripetal shifts:

$$T_i = \{p_j \mid (p_j + s_j) \in K_i\}. \tag{1}$$

**Label generation** The label generation for the probability map is inspired by PSENet [37], where the positive area of the text kernel (shaded areas in Fig. 3(b)) is generated by shrinking the annotated polygon (shaded areas in Fig. 3(a)) using the Vatti clipping algorithm [35]. The offset of shrinking is computed based on the perimeter and area of the original polygon and the shrinking ratio is set to 0.7 empirically. Since the annotations in the dataset may not perfectly fit the text instances well, we design a training mask $M$ to distinguish the supervision of the valid and ignoring regions. The text instance excluding the text kernel ($T_i - K_i$) is the ignoring region, which means that the gradients in this area are not propagated back to the network. The training mask $M$ can be formulated as follows:

$$M_j = \begin{cases} 0, & \text{if } p_j \in \bigcup_i (T_i - K_i) \\ 1, & \text{otherwise.} \end{cases} \tag{2}$$

We simply multiply the training mask by the loss of the segmentation branch to eliminate the influence brought by wrong annotations.

In the label generation step of the regression branch, the text instance affects the centripetal shift map in three ways. First, the centripetal shift in the background region ($I - \bigcup_i T_i$) should prevent the background pixels from entering into any text kernel and thus we set it to $(0, 0)$ intuitively. Second, the centripetal shift in the text kernel region ($\bigcup_i K_i$) should keep the locations of text kernel pixels unchanged and we also set it to $(0, 0)$ for convenience. Third, we expect that each pixel in the region of the text instance excluding the text kernel ($\bigcup_i (T_i - K_i)$) can be guided to its corresponding kernel by the centripetal shift. Therefore, we continuously conduct the erosion operation over the text kernel map twice, compare these two temporary results and obtain the text kernel reference (polygons with solid lines in Fig. 3(c)) as the destination of the generated centripetal shift. As shown in Fig. 3(d), we build the centripetal shift between each pixel in the shaded area and its nearest text kernel reference to prevent numerical accuracy issues caused by rounding off. Note that if two instances overlap, the

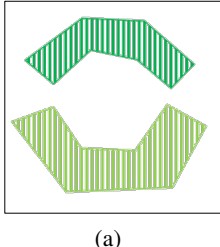 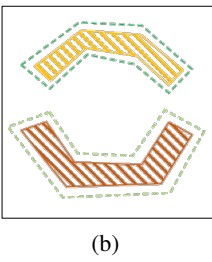 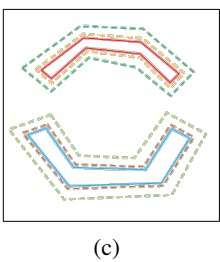 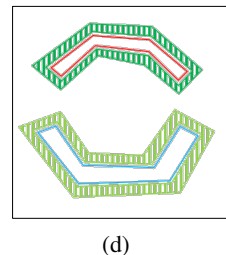

|  (a)  |  (b)  |  (c)  |  (d)  |

Figure 3: Label generation. (a) Text instance; (b) Text kernel; (c) Text kernel reference; (d) Text region that contains nonzero centripetal shifts.

smaller instance has the higher priority. Specifically, the centripetal shift $s_j$ can be calculated by:

$$
s_j = \begin{cases} (0,0), & \text{if } p_j \in (I - \bigcup_i T_i) \\ \overrightarrow{p_j p_j^*}, & \text{if } p_j \in \bigcup_i (T_i - K_i) \\ (0,0), & \text{if } p_j \in \bigcup_i K_i, \end{cases} \tag{3}
$$

where $p_j^*$ represents the nearest text kernel reference to the pixel $p_j$. During training, the Smooth L1 loss [5] is applied for supervision. Nevertheless, according to a previous observation [11], the dense regression can be modeled as a simple Dirac delta distribution, which fails to consider the ambiguity and uncertainty in the training data. To address the problem, we develop a regression mask $R$ for the relaxation operation and integrate it into the Smooth L1 loss to reduce the burden of model training. We extend the correct prediction from one specific value to a range and any centripetal shift which moves the pixel into the right region is treated as positive during training. The regression mask can be formulated as follows:

$$
R_j = \begin{cases} 0, & \text{if } p_j \in T_i \text{ and } (p_j + \widehat{s_j}) \in K_i \\ & \text{or } p_j \notin \bigcup_i T_i \text{ and } (p_j + \widehat{s_j}) \notin \bigcup_i K_i \\ 1, & \text{otherwise.} \end{cases} \tag{4}
$$

where $\widehat{s_j}$ and $s_j$ denote the predicted centripetal shift at the position $j$ and its ground truth, respectively. Like the segmentation loss, we multiply the regression mask by the Smooth L1 loss and form a novel loss function, namely the Relaxed L1 loss for dense centripetal shift prediction, to further improve the detection accuracy. The Relaxed L1 loss function can be formulated as follows:

$$
\mathcal{L}_{regression} = \sum_j \left( R_j \cdot \text{Smooth}_{\text{L1}}(s_j, \widehat{s_j}) \right), \tag{5}
$$

where $\text{Smooth}_{\text{L1}}()$ denotes the standard Smooth L1 loss.

### 3.2 Scene text detection with CentripetalText

**Network Architecture** In order to detect texts with arbitrary shapes fast and accurately, we adopt the efficient model design in PAN [38] and equip it with our CT and Relaxed L1 loss. First, ResNet18 [8] is used as the default backbone for fair comparison. Then, to remedy the weak representation ability of the lightweight backbone, two cascaded FPEMs [38] are utilized to continuously enhance the feature pyramid in both top-down and bottom-up manners. Afterwards, the generated feature pyramids of different depths are fused by FFM [38] into a single basic feature. Finally, we predict the probability map and the centripetal shift map from the basic feature for further contour generation.

**Inference** The procedure of text contour reconstruction is shown in Fig. 4. After feed-forwarding, the network produces the probability map and the centripetal shift map. We firstly binarize the probability map with a constant threshold (0.2) to get the binary map. Then, we find the connected components (text kernels) from the binary map as the clusters of pixel aggregation. Afterwards, we

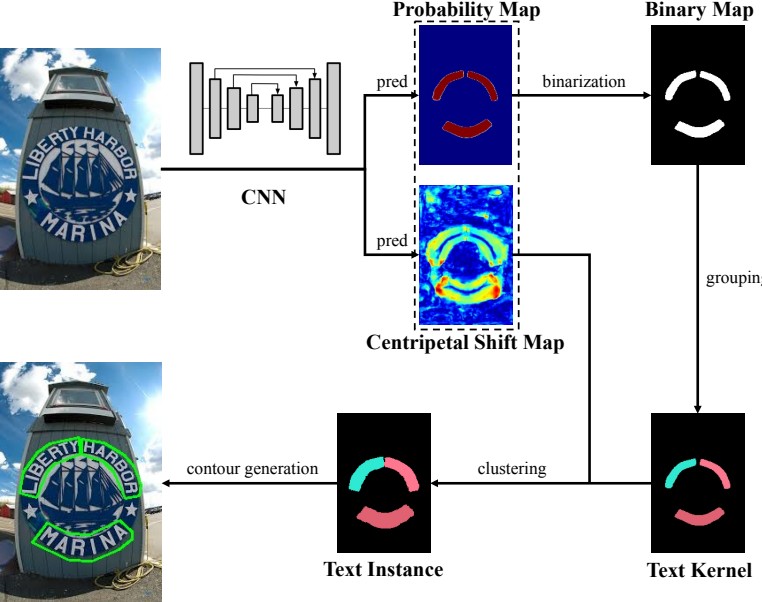

Figure 4: An overview of our proposed model.

assign each pixel to the corresponding cluster according to which text kernel (or background) can the pixel be shifted into by its centripetal shift. Finally, we build the text contour for each group of text pixels. Note that our post-processing strategy has an essential difference against PAN [38]. The post-processing in PAN is an iterative process, which gradually extends the text kernel to the text region by merging its neighbor pixels iteratively. On the contrary, we conduct the aggregation in one step, which means that the centripetal shifts of all pixels can be calculated in parallel by implementing one matrix operation, saving the inference time to a certain extent.

**Optimization**   Our loss function can be formulated as:

$$\mathcal{L} = \mathcal{L}_{segmentation} + \lambda\mathcal{L}_{regression}, \tag{6}$$

where $\mathcal{L}_{segmentation}$ denotes the segmentation loss of text kernels, and $\mathcal{L}_{regression}$ denotes the regression loss of centripetal shifts. $\lambda$ is a constant to balance the weights of the segmentation and regression losses. We set it to 0.05 in all experiments. Specifically, the prediction of text kernels is basically a pixel-wise binary classification problem and we apply the dice loss [26] to handle this task. Equipped with the training mask $M$, the segmentation loss can be defined as:

$$\mathcal{L}_{segmentation} = \sum_{j} \left( M_j \cdot \text{Dice}(c_j, \widehat{c}_j) \right), \tag{7}$$

where $\text{Dice}()$ denotes the dice loss function, $\widehat{c}_j$ and $c_j$ denote the predicted probability of text kernels at the position $j$ and its ground truth, respectively. Note that we adopt Online Hard Example Mining (OHEM) [32] to address the imbalance of positives and negatives while calculating $\mathcal{L}_{segmentation}$. Regarding the regression loss, a detailed description has been provided in Sec. 3.1.

**CentripetalText Proposal Network**   Our scene text detector is shrunk to a text proposal module, termed as CentripetalText Proposal Network (CPN), by transforming the polygonal outputs to the minimum area rectangles and instance masks. We follow the main design of the text detection and recognition modules of Mask TextSpotter v3 [13] and replace SPN with our CPN for the comparison of proposal quality and recognition accuracy.

# 4 Experiments

## 4.1 Datasets

**SynthText** [6] is a synthetic dataset, consisting of more than 800,000 synthetic images. This dataset is used to pre-train our model.

**Total-Text** [2] is a curved text dataset including 1,255 training images and 300 testing images. This dataset contains horizontal, multi-oriented, and curve text instances labeled at the word level.

**CTW1500** [47] is another curved text dataset including 1,000 training images and 500 testing images. The text instances are annotated at text-line level with 14-polygons.

**MSRA-TD500** [45] is a multi-oriented text dataset which contains 300 training images and 200 testing images with text-line level annotation. Due to its small scale, we follow the previous works [51, 23] to include 400 extra training images from HUST-TR400 [44].

## 4.2 Implementation details

To make fair comparisons, we use the same training settings described below. ResNet [8] pre-trained on ImageNet [4] is used as the backbone of our method. All models are optimized by the Adam optimizer with the batch size of 16 on 4 GPUs. We train our model under two training strategies: (1) learning from scratch; (2) fine-tuning models pre-trained on the SynthText dataset. Whichever training strategies, we pre-train models on SynthText for 50k iterations with a fixed learning rate of $1 \times 10^{-3}$, and train models on real datasets for 36k iterations with the "poly" learning rate strategy [50], where "power" is set to 0.9 and the initial learning rate is $1 \times 10^{-3}$. We follow the official implementation of PAN to implement data augmentation, including random scale, random horizontal flip, random rotation, and random crop. The blurred texts labeled as DO NOT CARE are ignored during training. In addition, we set the negative-positive ratio of OHEM to 3, and the shrinking rate of text kernel to 0.7. All those models are tested with a batch size of 1 on a GTX 1080Ti GPU without bells and whistles. For the task of end-to-end scene text recognition, we leave the original training and testing settings of Mask TextSpotter v3 unchanged.

## 4.3 Ablation study

To analyze our designs in depth, we conduct a series of ablation studies on both curve and multi-oriented text datasets (Total-Text and MSRA-TD500). In addition, all models in this subsection are trained from scratch.

Table 1: Quantitative results of our text detection models with different backbones and necks. "F" means F-measure.

| Backbone | Neck | Total-Text | | MSRA-TD500 | |
|---|---|---|---|---|---|
| | | F | FPS | F | FPS |
| ResNet18 | FPN [18] | 84.0 | **46.8** | 81.6 | **42.0** |
| | FPEM [38] | 84.9 | 40.0 | 83.0 | 34.8 |
| ResNet50 | FPN [18] | 85.1 | 25.5 | 82.7 | 21.9 |
| | FPEM [38] | **85.6** | 24.0 | **83.5** | 20.5 |

Table 2: Comparison between PAN [38] and our models with different regression losses. "Rep." denotes representation and "F" means F-measure.

| Rep. | Regression Loss | Total-Text | | MSRA-TD500 | |
|---|---|---|---|---|---|
| | | F | FPS | F | FPS |
| PAN [38] | | 83.5 | 39.6 | 78.9 | 30.2 |
| CT (Ours) | Smooth L1[5] | 83.0 | **40.0** | 81.0 | **34.8** |
| | Balanced L1[27] | 83.8 | **40.0** | 81.7 | **34.8** |
| | Relaxed L1 | **84.9** | **40.0** | **83.0** | **34.8** |

On the one hand, to make full use of the capability of the proposed CT, we try different backbones and necks to find the best network architecture. As shown in Tab. 1, although "ResNet18 + FPN" and "ResNet50 + FPEM" are the fastest and most accurate detectors, respectively, "ResNet18 + FPEM" achieves the best trade-off between accuracy and speed. Thus, we keep this combination by default in the following experiments. On the other hand, we study the validity of the Relaxed L1 loss by replacing it with others. Compared with the baseline Smooth L1 Loss [5] and the newly-released Balanced L1 loss [27], the F-measure value of our method improves over 1% on both two datasets, which indicates the effectiveness of the Relaxed L1 loss. Moreover, under the same settings of the model architecture, our method outperforms PAN by a wide extent while keeping its fast inference speed, indicating that the proposed CT is more efficient.

Table 3: Quantitative detection results on Total-Text and CTW1500. "P", "R" and "F" represent the precision, recall, and F-measure, respectively. "Ext." denotes external training data. * indicates the multi-scale testing is performed.

| Method | Ext. | Venue | Total-Text | | | | CTW1500 | | | |
|---|---|---|---|---|---|---|---|---|---|---|
| | | | P | R | F | FPS | P | R | F | FPS |
| CTPN [34] | - | ECCV'16 | - | - | - | - | 60.4 | 53.8 | 56.9 | 7.1 |
| SegLink [31] | - | CVPR'17 | 30.3 | 23.8 | 26.7 | - | 42.3 | 40.0 | 40.8 | 10.7 |
| EAST [51] | - | CVPR'17 | 50.0 | 36.2 | 42.0 | - | 78.7 | 49.1 | 60.4 | 21.2 |
| PSENet [37] | - | CVPR'19 | 81.8 | 75.1 | 78.3 | 3.9 | 80.6 | 75.6 | 78.0 | 3.9 |
| PAN [38] | - | ICCV'19 | 88.0 | 79.4 | 83.5 | 39.6 | 84.6 | 77.7 | 81.0 | 39.8 |
| **CT-320** | - | - | 87.6 | 72.7 | 79.4 | **93.2** | **85.7** | 73.2 | 79.0 | **107.2** |
| **CT-512** | - | - | 87.9 | 80.8 | 84.2 | 57.0 | 85.2 | 78.4 | 81.7 | 59.8 |
| **CT-640** | - | - | **88.8** | **81.4** | **84.9** | 40.0 | 85.5 | **79.2** | **82.2** | 40.8 |
| TextSnake [23] | ✓ | ECCV'18 | 82.7 | 74.5 | 78.4 | - | 67.9 | **85.3** | 75.6 | - |
| MSR [43] | ✓ | IJCAI'19 | 83.8 | 74.8 | 79.0 | - | 85.0 | 78.3 | 81.5 | - |
| SegLink++ [33] | ✓ | PR'19 | 82.1 | 80.9 | 81.5 | - | 82.8 | 79.8 | 81.3 | - |
| PSENet [37] | ✓ | CVPR'19 | 84.0 | 78.0 | 80.9 | 3.9 | 84.8 | 79.7 | 82.2 | 3.9 |
| SPCNet [40] | ✓ | AAAI'19 | 83.0 | 82.8 | 82.9 | - | - | - | - | - |
| LOMO* [48] | ✓ | CVPR'19 | 87.6 | 79.3 | 83.3 | - | 85.7 | 76.5 | 80.8 | - |
| CRAFT [1] | ✓ | CVPR'19 | 87.6 | 79.9 | 83.6 | - | 86.0 | 81.1 | 83.5 | - |
| Boundary [36] | ✓ | AAAI'20 | 85.2 | 83.5 | 84.3 | - | - | - | - | - |
| DB [16] | ✓ | AAAI'20 | 87.1 | 82.5 | 84.7 | 32.0 | 86.9 | 80.2 | 83.4 | 22.0 |
| PAN [38] | ✓ | ICCV'19 | 89.3 | 81.0 | 85.0 | 39.6 | 86.4 | 81.2 | 83.7 | 39.8 |
| DRRG [49] | ✓ | CVPR'20 | 86.5 | **84.9** | 85.7 | - | 85.9 | 83.0 | **84.5** | - |
| **CT-320** | ✓ | - | 88.0 | 75.4 | 81.2 | **93.2** | 87.7 | 74.7 | 80.7 | **107.2** |
| **CT-512** | ✓ | - | 90.2 | 81.5 | 85.6 | 57.0 | 87.8 | 79.0 | 83.2 | 59.8 |
| **CT-640** | ✓ | - | **90.5** | 82.5 | **86.3** | 40.0 | **88.3** | 79.9 | 83.9 | 40.8 |

## 4.4 Comparisons with state-of-the-art methods

**Curved text detection** We first evaluate our CT on the Total-Text and CTW1500 datasets to examine its capability for curved text detection. During testing, we set the short side of images to different scales (320, 512, 640) and keep their aspect ratios. We compare our methods with other state-of-the-art detectors in Tab. 3. For Total-Text, when learning from scratch, CT-640 achieves the competitive F-measure of 84.9%, surpassing most existing methods pre-trained on external text datasets. When pre-training on SynthText, the F-measure value of our best model CT-640 reaches 86.3%, which is 0.6% better than second-best DRRG [49], while still ensuring the real-time detection speed (40.0 FPS). Fig. 1 demonstrates the accuracy-speed trade-off of some top-performing real-time text detectors, from which it can be observed that our CT breaks through the limitation of accuracy-speed boundary. Analogous results can also be obtained on CTW1500. With external training data, the F-measure of CT-640 is 83.9%, the second place of all methods, which is only lower than DGGR. Meanwhile, the speed can still exceed 40 FPS. In summary, the experiments conducted on Total-Text and CTW1500 demonstrate that the proposed CT achieves superior or competitive results compared to state-of-the-art methods, indicating its superiority in modeling curved texts. We visualize our detection results in Fig. 5 for further inspection.

Table 4: Quantitative detection results on MSRA-TD500. "P", "R" and "F" represent the precision, recall, and F-measure, respectively. "Ext." denotes external training data.

| Method | Ext. | P | R | F | FPS |
|---|---|---|---|---|---|
| RRPN [25] | - | 82.0 | 68.0 | 74.0 | - |
| EAST [51] | - | **87.3** | 67.4 | 76.1 | 13.2 |
| PAN [38] | - | 80.7 | 77.3 | 78.9 | 30.2 |
| **CT-736** | - | 87.1 | **79.3** | **83.0** | 34.8 |
| SegLink [31] | ✓ | 86.0 | 70.0 | 77.0 | 8.9 |
| PixelLink [3] | ✓ | 83.0 | 73.2 | 77.8 | 3.0 |
| TextSnake [23] | ✓ | 83.2 | 73.9 | 78.3 | 1.1 |
| RRD [17] | ✓ | 87.0 | 73.0 | 79.0 | 10.0 |
| TextField [42] | ✓ | 87.4 | 75.9 | 81.3 | - |
| CRAFT [1] | ✓ | 88.2 | 78.2 | 82.9 | 8.6 |
| MCN [21] | ✓ | 88.0 | 79.0 | 83.0 | - |
| PAN [38] | ✓ | 84.4 | **83.8** | 84.1 | 30.2 |
| DB [16] | ✓ | **91.5** | 79.2 | 84.9 | 32.0 |
| DRRG [49] | ✓ | 88.1 | 82.3 | 85.1 | - |
| **CT-736** | ✓ | 90.0 | 82.5 | **86.1** | 34.8 |

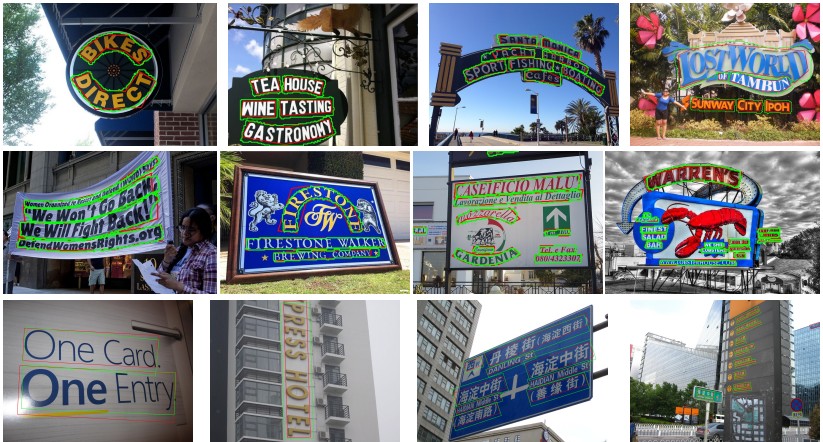

Figure 5: Qualitative results of the proposed method. Images in row 1-3 are sampled from Total-Text, CTW1500, and MSRA-TD500, respectively. Ground-truth annotations are in red and our detection results are in green.

**Multi-oriented text detection**   We also evaluate CT on the MSRA-TD500 dataset to test the robustness in modeling multi-oriented texts. As shown in Tab. 4, CT achieves the F-measure value of 83.0% at 34.8 FPS without external training data. Compared with PAN, our method outperforms it by 4.1%. When pre-training on SynthText, the F-measure value of our CT can further be boosted to 86.1%. The highest performance and the fastest speed achieved by CT prove its generalization ability to deal with texts with extreme aspect ratios and various orientations in complex natural scenarios.

Table 5: Quantitative end-to-end recognition results on Total-Text. The evaluation protocol is the same as the one in Mask TextSpotter v3 [13]. "None" means recognition without any lexicon. "Full" lexicon contains all words in the test set. * indicates the multi-scale testing is performed.

| Method | None | Full |
|---|---|---|
| TextBoxes* [15] | 36.3 | 48.9 |
| Mask TextSpotter v1 [24] | 52.9 | 71.8 |
| Qin et al. [28] | 63.9 | - |
| Boundary [36] | 65.0 | 76.1 |
| Mask TextSpotter v2 [12] | 65.3 | 77.4 |
| CharNet* [41] | 69.2 | - |
| ABCNet* [20] | 69.5 | 78.4 |
| Mask TextSpotter v3 [13] | 71.2 | 78.4 |
| **Mask TextSpotter v3 w/ CPN** | **71.9** | **79.5** |

**End-to-end text recognition**   We simply replace SPN in Mask TextSpotter v3 with our proposed CPN to develop a more powerful end-to-end text recognizer. We evaluate CPN-based text spotter on Total-Text to test the proposal generation quality for the text spotting task. As shown in Tab. 5, equipped with CPN, Mask TextSpotter v3 achieves the F-measure values of 71.9% and 79.5% when the lexicon is not used and used respectively. Compared with the original version and other state-of-the-art methods, our method can obtain higher performance whether the lexicon is provided or not. Thus, the quantitative results demonstrate that CPN can produce more accurate text proposals than SPN, which is beneficial for recognition and can improve the performance of end-to-end text recognition further.

We visualize the text proposals and the polygon masks generated by SPN and CPN for intuitive comparison. As shown in Fig. 6, we can see that the polygon masks produced by CPN fit the text instances more tightly, which qualitatively proves the superiority of the proposed CPN in producing text proposals compared to other approaches.

## 5   Conclusion

To keep the simplicity and robustness of text instance representation, we proposed CentripetalText (CT) which decomposes text instances into the combination of text kernels and centripetal shifts. Text kernels identify the skeletons of text instances while centripetal shifts guide the external text pixels to the internal text kernels. Moreover, to reduce the burden of model training, a relaxation operation was integrated into the dense regression for centripetal shifts, allowing the correct prediction in a

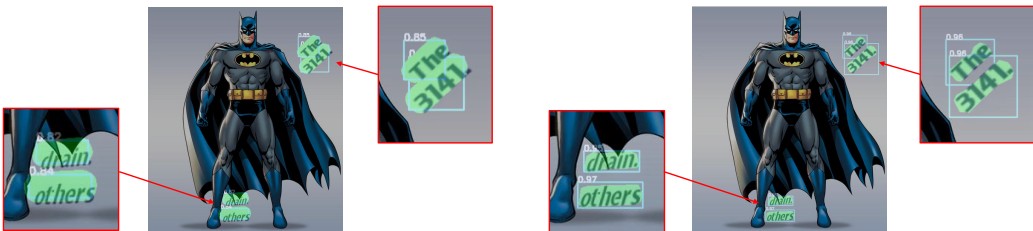

(a) Segmentation Proposal Network (SPN)    (b) CentripetalText Proposal Network (CPN)

Figure 6: Qualitative comparison of proposals obtained by SPN and CPN. The blue rectangles denote the text proposals and the green areas denote the binary polygon masks.

range. Equipped with the proposed CT, our detector achieved superior or comparable performance compared to other state-of-the-art methods while keeping the real-time inference speed. The source code is available at https://github.com/shengtao96/CentripetalText, and we hope that the proposed CT can serve as a valuable and common representation for scene texts.

## Acknowledgments and Disclosure of Funding

This work was supported by Beijing Nova Program of Science and Technology (Grant No.: Z191100001119077), Project 2020BD020 supported by PKU-Baidu Fund, National Language Committee of China (Grant No.: ZDI135-130), Center For Chinese Font Design and Research, Key Laboratory of Science, Technology and Standard in Press Industry (Key Laboratory of Intelligent Press Media Technology), State Key Laboratory of Media Convergence Production Technology and Systems.

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
