# A    Rotation robustness analysis

Table 6: Quantitative end-to-end recognition results (without lexicon) on Rotated ICDAR2013. The evaluation protocol is the same as the one in ICDAR2015 dataset. CharNet is tested with the official released model. Mask TextSpotter v2 (MTSv2), Mask TextSpotter v3 (MTSv3) and our model (MTSv3 w/ CPN) are trained with the same rotating augmentation. "RA" is short for rotating angles. "P", "R" and "F" represent the precision, recall and F-measure respectively.

| RA(°) | CharNet [41] | | | MTSv2 [12] | | | MTSv3 [13] | | | MTSv3 w/ CPN | | |
|---|---|---|---|---|---|---|---|---|---|---|---|---|
| | P | R | F | P | R | F | P | R | F | P | R | F |
| 0 | 61.7 | 61.2 | 61.4 | 86.3 | 75.2 | 80.3 | 89.0 | 73.0 | 80.2 | 89.7 | 76.3 | 82.4 |
| 15 | 66.3 | 61.9 | 64.0 | 78.4 | 53.5 | 63.6 | 87.2 | 69.8 | 77.5 | 87.8 | 72.0 | 79.1 |
| 30 | 60.9 | 56.5 | 58.6 | 73.9 | 54.7 | 62.9 | 87.8 | 67.5 | 76.3 | 89.6 | 69.4 | 78.2 |
| 45 | 34.2 | 33.5 | 33.9 | 66.4 | 45.8 | 54.2 | 88.5 | 66.8 | 76.1 | 89.4 | 66.9 | 76.5 |
| 60 | 10.3 | 8.4 | 9.3 | 68.2 | 48.3 | 56.6 | 88.5 | 67.6 | 76.6 | 88.6 | 67.1 | 76.4 |
| 75 | 0.3 | 0.2 | 0.2 | 77.0 | 59.2 | 67.0 | 86.9 | 67.6 | 76.0 | 88.1 | 67.7 | 76.5 |
| 90 | 0.0 | 0.0 | 0.0 | 82.0 | 56.9 | 67.1 | 85.9 | 57.9 | 69.1 | 87.8 | 60.6 | 71.7 |

To further demonstrate the rotation robustness of our method, we evaluate our CPN-based text spotter on the Rotated ICDAR2013 dataset.

**Rotated ICDAR2013** [13] is an augmented text dataset that is generated from ICDAR2013 [10]. To form the Rotated ICDAR2013 dataset, all the images and annotations in the test set of the ICDAR2013 benchmark are rotated with some specific angles, including 15°, 30°, 45°, 60°, 75° and 90°. The dataset contains 229 training images and 233 testing images. The text instances are annotated at the text-line level with rotated rectangles. Since the annotations are extended from horizontal rectangles to multi-oriented ones, we adopt the evaluation protocols in the ICDAR2015 dataset [9].

As shown in Tab. 6, we compare three top-performing methods CharNet [41], Mask TextSpotter v2 [12], and Mask TextSpotter v3 [13] with our proposed text spotter at different rotation angles. We can see that CharNet and Mask TextSpotter v2 fail to deal with the multi-oriented texts and their performances fall well below ours. Moreover, Our method surpasses Mask TextSpotter v3 by more than 1.5% when the rotation angles are 0°, 15°, 30° and 90°, and we obtain the competitive performance under the other angles. The extensive experiments prove the superior robustness to various orientations of scene texts offered by our method.

# B    Limitation

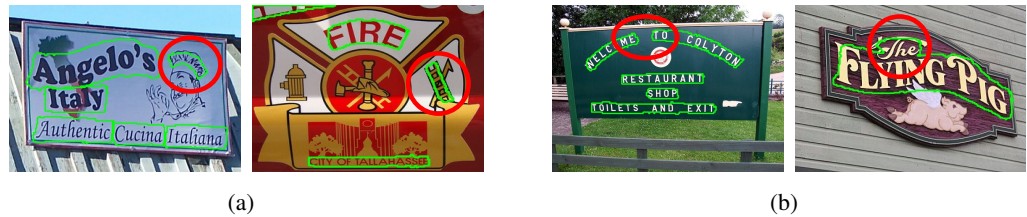

(a)                                          (b)

Figure 7: Failure samples.

Although the proposal CT works well in most cases of scene text detection, it still fails in some difficult cases as shown in Fig. 7. On the one hand, our method may mistakenly treat some decorative patterns as texts and thus produces false positives (see Fig. 7(a)). In this situation, the followed recognition module can effectively restrain such failures according to the high-level semantic information. On the other hand, whether two close text instance should be connected into one or not is still a challenging problem which influences the detection performance deeply (see Fig. 7(b)). In the future, we plan to solve this problem and make the model more robust.

## C More detection and recognition results

More detection results are shown in Fig. 8 (Total-Text), Fig. 9 (CTW1500), and Fig. 10 (MSRA-TD500), and end-to-end recognition results on Total-Text are shown in Fig. 11.

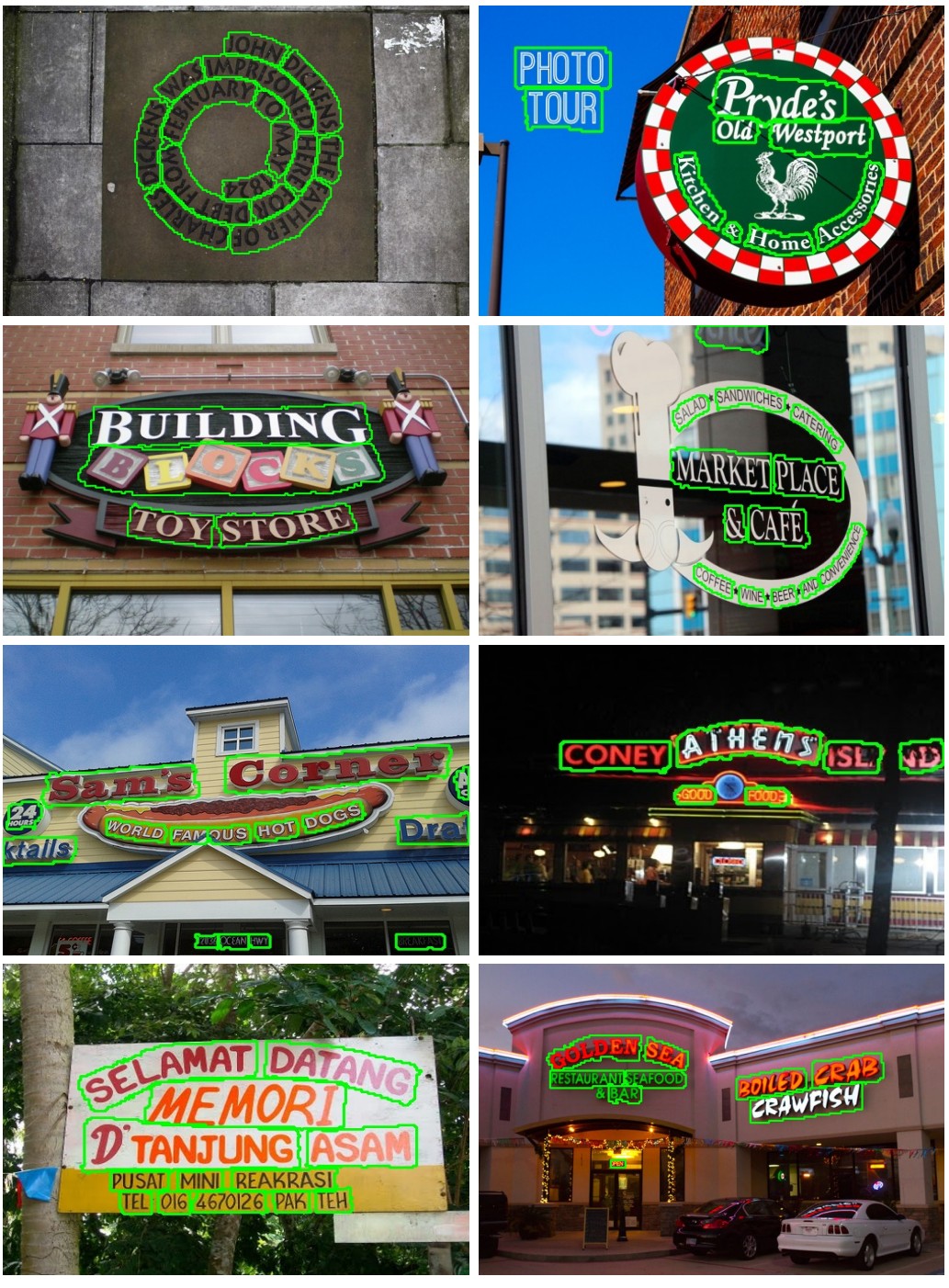

Figure 8: Detection results on Total-Text.

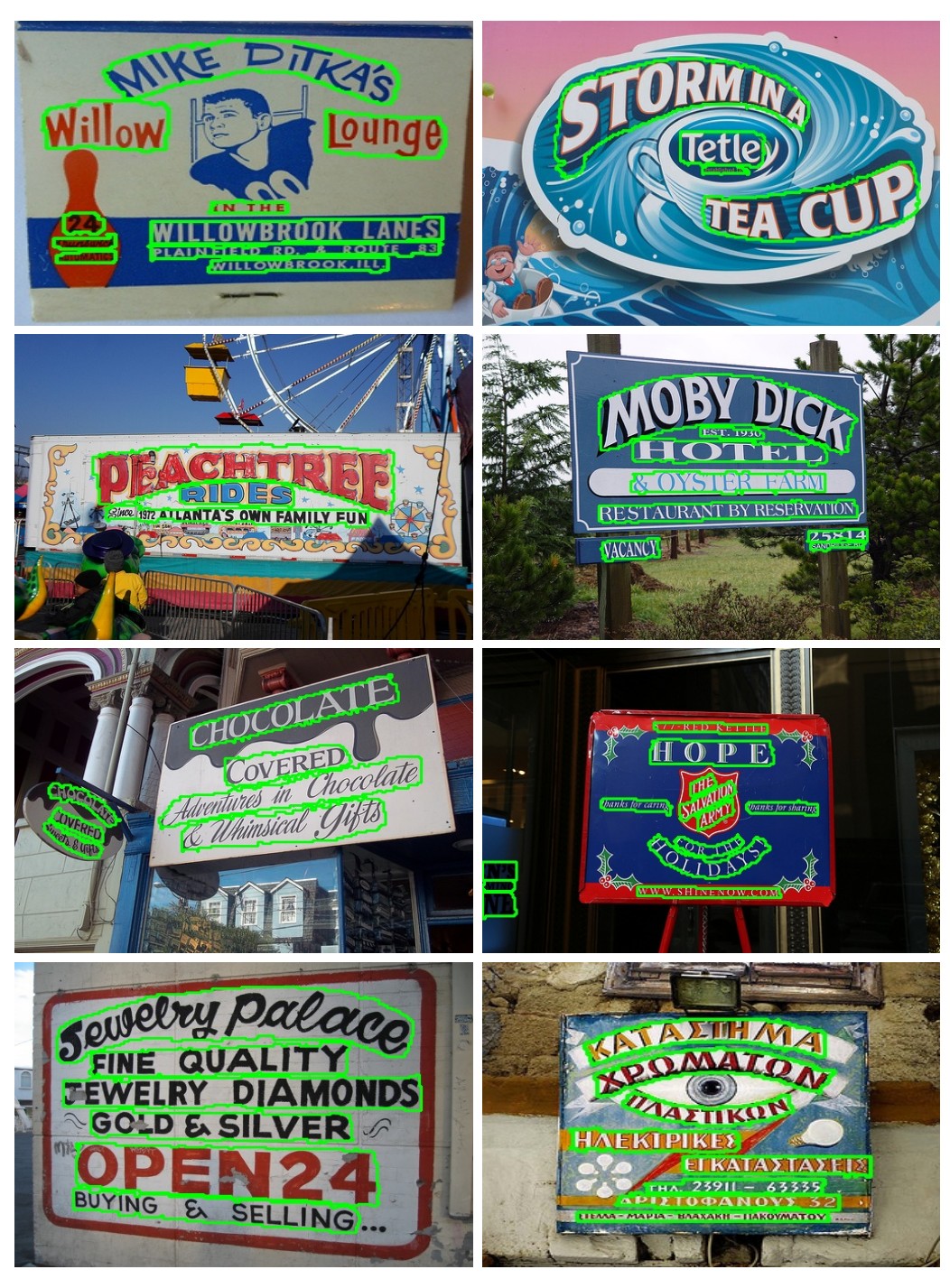

Figure 9: Detection results on CTW1500.

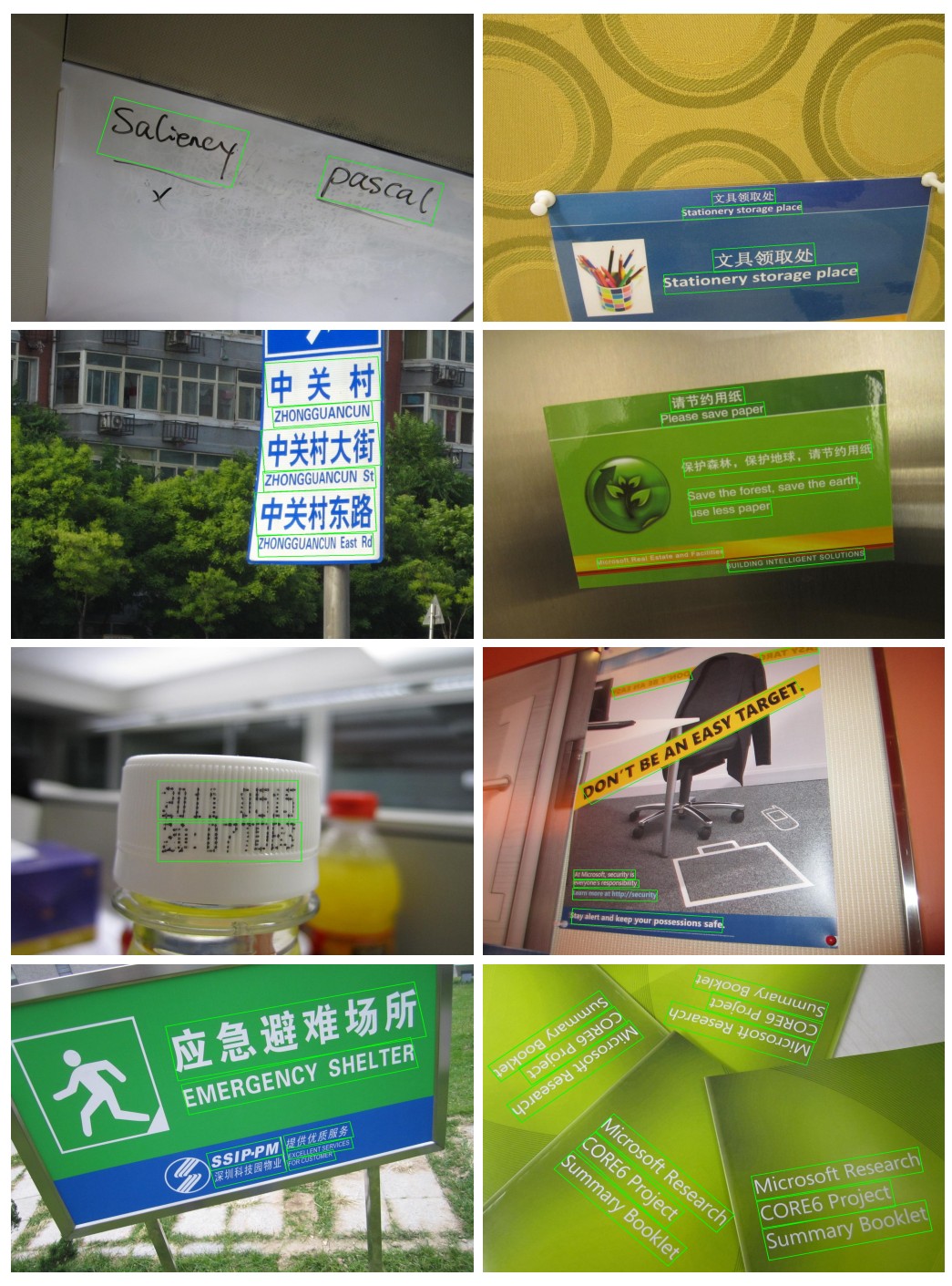

Figure 10: Detection results on MSRA-TD500.

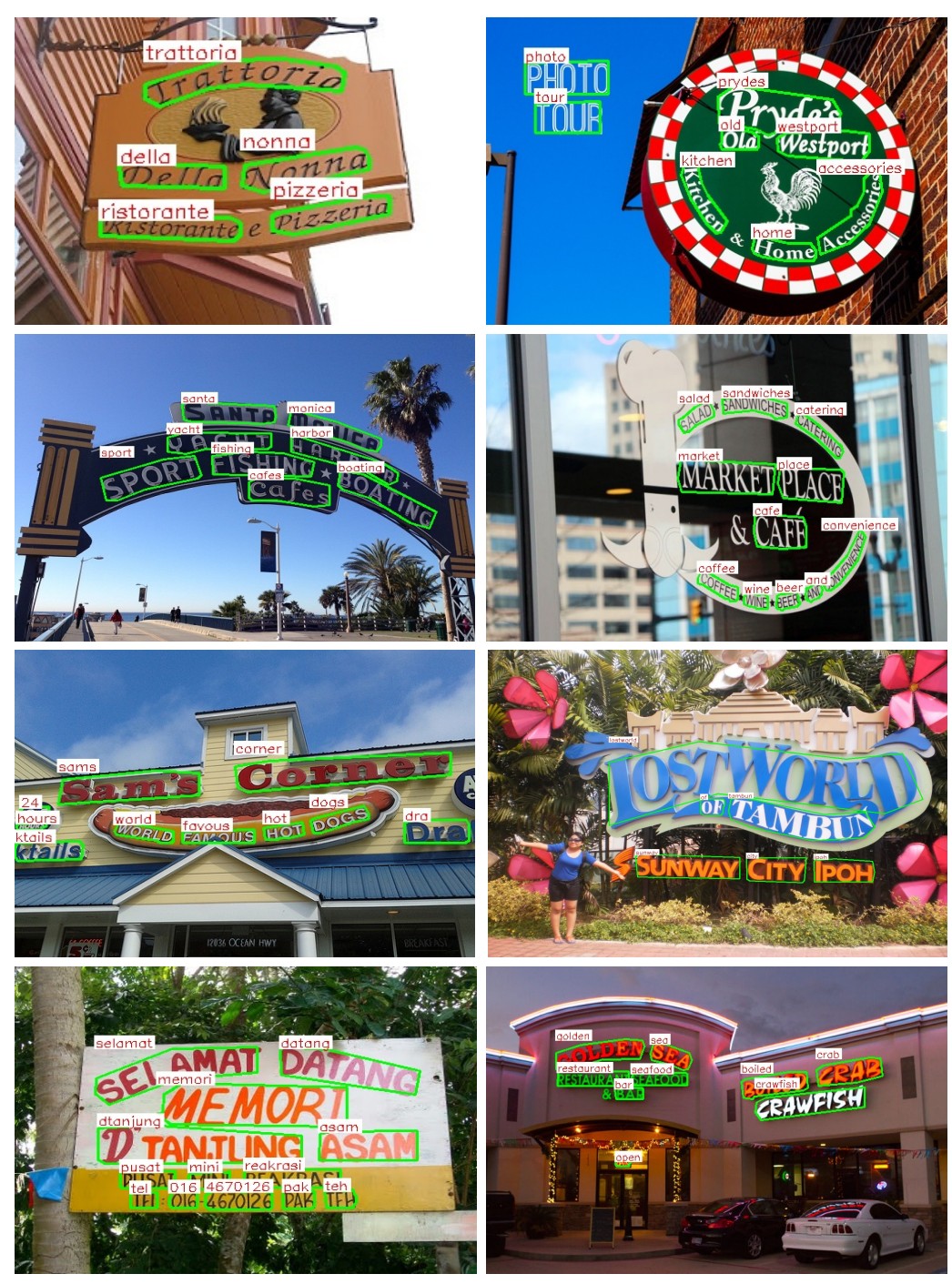

Figure 11: Recognition results on Total-Text.