# OpenReview forum: "CentripetalText: An Efficient Text Instance Representation for Scene Text Detection"
_NeurIPS.cc/2021/Conference — NeurIPS 2021 Poster_

### Official Review · Reviewer_CRQN · 2021-07-15

**Rating:** 8
**Confidence:** 4

**Summary:**

Update after discussion among reviewers: I am still strongly in favor of accepting this paper.

The paper presents a new method to represent arbitrary shaped text in images and shows how this representation can be used in the output layer of some of the recent text detection methods.

The results look qualitatively and quantitative great.

The method uses a simple representation that is intuitively understandable.



**Ethical Concerns:**

it's a standard domain - I feel it's an interesting question to think about the general societal impact of scene text recognition - but I wouldn't think this needs to be discussed in this paper which provides an elegant, yet incremental improvement to previous methods.

**Limitations And Societal Impact:**

it's a standard domain - I feel it's an interesting question to think about the general societal impact of scene text recognition - but I wouldn't think this needs to be discussed in this paper which provides an elegant, yet incremental improvement to previous methods.

**Main Review:**

My biggest concern with the paper is that there appears to be a mismatch between equations 2 and 3:

in eq. 2 the loss for the region \bigcup(T_i-K_i) is masked out.
in eq. 3 this is however the most important region.

I am not sure how the method is able to learn anything about this region if the loss is fully masked.


The paper does provide a great comparison to many standard methods.

The method is well motivated - and I like how it can be applied in the output layer of a bunch of different networks.


minor comments:
abstract:
most intractable -> hardest
abbreviation SPN is not explained.
1.1% improvement (relative or absolute?)

generally, the paper should be carefully copy read and some typos and grammatical mistakes could be fixed.
some examples:
line 49: the mistakes -> mistakes
line 57: by the CT -> by CT
line 58: oppoonents -> methods
captuion fig 2: helpless -> useless

beyond that: the introduction and the abstract feel rather repetitive.

**Time Spent Reviewing:**

2

---

> ### Author Response · Authors · 2021-08-10
> **Responses to the Comments of Reviewer CRQN**
>
> Thank you so much for your strong support and constructive comments. Below are our detailed responses to your concerns:
>
> Comment_1:
>
> “My biggest concern with the paper is that there appears to be a mismatch between equations 2 and 3:
> in eq. 2 the loss for the region \bigcup(T_i-K_i) is masked out. in eq. 3 this is however the most important region.
> I am not sure how the method is able to learn anything about this region if the loss is fully masked.”
>
> Response_1:
>
> Thanks for giving us the opportunity to explain your concern at this point. Actually, Eq. 2 and Eq. 3 are relatively independent. They will not be multiplied together when computing the final loss during training. More specifically, the final loss contains two parts (Eq. 6), the segmentation branch and the regression branch (used for predicting centripetal shift values), which are computed by Eq. 7 and Eq. 5 respectively. $M_j$ (Eq. 2) is the mask of the segmentation branch for better text kernel prediction, while $s_j$ (Eq. 3) is used for supervising the regression branch for centripetal shifts construction. In fact, $R_j$ (Eq. 4) is the mask of the regression branch, which is multiplied by $s_j$ at last. Therefore, $M_j$ (Eq. 2) will not mask out the regression loss for the region $\bigcup\limits_i(T_i-K_i)$ in Eq. 3.
>
> Comment_2:
>
> “minor comments: abstract: most intractable -> hardest abbreviation SPN is not explained. 1.1% improvement (relative or absolute?)”
>
> Response_2:
>
> Nice suggestions. We will replace “most intractable” with “hardest” in abstract.
>
> “SPN” refers to Segmentation Proposal Network, which was originally proposed in Mask TextSpotter v3 [11]. We will add its full name in the place where it appears for the first time.
>
> 1.1% improvement is relative to the result (78.4%) of Mask TextSpotter v3 on Total-text , which can be obtained by comparing the last two rows in Tab. 5. We will revise the expression to ease the probable misunderstanding.
>
> Comment_3:
>
> “generally, the paper should be carefully copy read and some typos and grammatical mistakes could be fixed. some examples: line 49: the mistakes -> mistakes line 57: by the CT -> by CT line 58: oppoonents -> methods captuion fig 2: helpless -> useless
> beyond that: the introduction and the abstract feel rather repetitive.”
>
> Response_3:
>
> We will carefully check the whole manuscript and ensure that all typos and grammatical mistakes will be fixed. The abstract will be rewritten to prevent repeated descriptions in these two sections.

---

### Official Review · Reviewer_ZsrE · 2021-07-16

**Rating:** 6
**Confidence:** 4

**Summary:**

This paper proposes a novel representation of text instances and an algorithm that detects text instances in an image. A text instance is represented as a kernel that is a shrunk version of the text instance along with surrounding pixels that belong to the kernel based on centripetal shifts. This representation simplifies the post-processing to obtain text instances from the output of neural network but yet yields accurate detection. The experimental results confirm the effectiveness of the approach on common scene text datasets containing irregular texts.

**Ethical Concerns:**

No.

**Limitations And Societal Impact:**

Limitations are not particularly discussed in the paper.

No concerns on potential negative societal impact.

**Main Review:**

The proposed method successfully simplifies the post-processing to obtain the final results from the output of neural network while keeping the flexibility of the pixel-level representation. The paper proposes a solution that solves the problems raised in the paper. Given the simplicity and the effectiveness, it could be one of the representations widely used for scene text recognition. It seems to have good novelty and originality.

However, the paper seems to need more work for accept. The followings are my main concerns on the paper:


1. The text explaining the Relaxed L1 loss seems to contain errors or very confusing. I was not able to read lines 167-178 very well. Especially, Eq. (4) and (5) did not make sense to me. I might be just wrong, but it looks like R_j is 0 everywhere according to Eq. (1). Then, Eq. (5) is just 0. I would want to be corrected if I do not understand the equations right. Some visualization might help.


2. The comparisons with previous methods (for text representation) are conducted by only referring to the numbers reported in the original papers. Having such comparisons are still useful, but they do not necessarily help to demonstrate the superiority of the proposed method due to inevitable differences in their experimental conditions. It would be necessary to re-implement some of the methods that suffer from the problems (i.e. ones losing either simplicity or robustness) and compare the methods in a truly fair manner. If it is challenging to re-implement existing methods, it would be at least necessary to detail the differences among methods and explain the reasons why it is believed that the proposed method is superior.


3. I know that it is unfortunately common in the domain, but I am not sure how much informative it is to compare FPS among methods where each experiment could be done in a different condition. I think it is OK to list those numbers, but I am afraid that it does not prove the superiority of the proposed method.


4. The writing could be improved. Sections 1-3 are entangled and could be cleaned up. For example, Fig. 3 (or the overview of the method) is not explained in Section 3 but in Section 1. Section 2 is too short and important relevant work like [35] is mentioned only in Section 1.


5. Should c_j at line 140 be s_j ?

[update after author response]

My biggest concern (1) was addressed by the author response.

As also pointed out by reviewer U3is, we want to expect more detailed analyses of the proposed method. That will include more ablation studies, comparisons with locally-implemented appropriate strong baselines for the method, discussions on "alternative (not) considered", etc.

Although more analyses are demanded, I see no major flaw in the paper now and it will have a certain value to the community.

I'm happy to correct my rating to 6.

**Time Spent Reviewing:**

4

---

> ### Author Response · Authors · 2021-08-10
> **Responses to the Comments of Reviewer ZsrE**
>
> Thanks for your constructive comments. Below are our detailed responses to your concerns:
>
> Comment_1:
>
> “The proposed method successfully simplifies the post-processing to ……. It seems to have good novelty and originality.
> However, the paper seems to need more work for accept. The followings are my main concerns on the paper:”
>
> Response_1:
>
> Thanks for your support on our paper by mentioning “Given the simplicity and the effectiveness, it could be one of the representations widely used for scene text recognition”. Also, thanks for your appreciation on the novelty and originality of our paper by saying “It seems to have good novelty and originality.” We will carefully follow your constructive suggestions to revise the paper to make it be a more qualified NeurIPS paper. Detailed responses to your comments are listed as follows.
>
> Comment_2:
>
> “1. The text explaining the Relaxed L1 loss seems to …... I would want to be corrected if I do not understand the equations right. Some visualization might help.”
>
> Response_2:
>
> We believe that you might have some misunderstandings regarding the Relaxed L1 loss. Thank you for giving us the valuable opportunity for explanation. Eq. (4) and (5) are both reasonably designed. Since the regression mask is mainly used for training, the gradients of pixels that have been correctly predicted are set to 0 according to Eq. (4) during training. Only the gradients of incorrectly predicted pixels are used to update the parameters of our neural network. Thereby, $R_j$ will be 0 everywhere only when all pixels have been correctly classified. In other words, we simply set a threshold to judge whether the label of each pixel has been correctly predicted or not instead of forcing it to have the same centripetal shift value as the ground truth. As also suggested by Reviewer U3is, we will add a figure to show “an example for the regression mask and its usefulness” to let readers “better understand the motivation”.
>
> Comment_3:
>
> “2. The comparisons with previous methods (for text representation) are conducted by only referring to the numbers reported in the original papers. ……. and explain the reasons why it is believed that the proposed method is superior.”
>
> Response_3:
>
> Actually, we have re-implemented most of the methods compared in our experiments as long as their source codes are available. For some methods that do not have source codes publicly available and are also hard to be reproduced, we have to directly list the numbers reported in the original papers. This is acceptable and commonly used in the literature. Since most of those methods compared in this paper were implemented by us under the same condition and the GPU adopted here is NVIDIA 1080ti that has been widely used for several years, we believe that the conclusion drawn based on our experimental results is considerably reasonable.
>
> For a more intuitive understanding of our method, here we detail some differences between our work with PAN[35] and MSR[40]:
>
> Comparison with PAN[35]. Our method uses centripetal shifts instead of embeddings to guide the text pixel aggregation. Empirically, centripetal shifts are easier to learn than embeddings, because centripetal shifts have physical meanings and it is difficult to make the embeddings of the text pixels (including some background pixels caused by inadequate labeling) the same. The better convergence brings better accuracy and the relaxation operation in the Relaxed L1 Loss further expands it. What's more, our post-processing is conducted in one step (calculating the shifted position and judging whether it is in some text kernel), replacing the iterative steps (aggregating nearby pixels step by step) in PAN[35], which guarantees our fast speed.
>
> Comparison with MSR[40]. Although we all output text kernels and shifts, our method is essentially a pixel clustering which leads the text pixels to the text kernels according to centripetal shifts, while MSR[40] predicts text boundary points and then uses geometric algorithms to rebuild contours, which consumes more time at the post-processing stage. What's more, our centripetal shifts are from (external) text pixels to (internal) text kernel pixels, while the shifts in MSR[40] are from (internal) text kernel pixels to (external) text boundary points. Intuitively, the prediction in our manner will be easier. To the best of our knowledge, our paper is the first text detection work that implements the prediction from external pixels to internal pixels.
>
> Comment_4:
>
> “3. I know that it is unfortunately common in the domain, ……. but I am afraid that it does not prove the superiority of the proposed method.”
>
> Response_4:
>
> As mentioned above, most of those representative methods (i.e., PAN, DB, Mask TextSpotter v3, etc.) have been re-implemented by us using the official source codes released by corresponding authors under the same experimental condition. Furthermore, the GPU we used in our experiments is NVIDIA 1080ti which has been widely used for many years. For those methods we did not re-implement, most of them were tested with the similar or better GPUs, and thus similar or smaller FPS numbers could be obtained under our condition. Therefore, the superiority of our method considering the balance of detection accuracy and speed as shown in Figure 1 can be proved by our experimental results.
>
> Comment_5:
>
> “4. The writing could be improved. Sections 1-3 are entangled and could be cleaned up.…...”
>
> Response_5:
>
> Thanks for your valuable suggestions. As we know, the introduction section needs to explain the motivation of the proposed method clearly and thus it is inevitable that some of the contents in the sections of related work and method description (i.e., Section 2&3) will be entangled with the introduction section. Anyway, we will try our best to disentangle these three sections by following your suggestions. Thanks again!
>
> Comment_6:
>
> “5. Should c_j at line 140 be s_j ?”
>
> Response_6:
>
> Yes, we made a mistake here. The c_j at line 140 should be replaced by s_j. Thanks!
>
> Comment_7:
>
> “Limitations are not particularly discussed in the paper.”
>
> Response_7:
>
> We have provided a discussion in the supplemental material (see Figure 7 and the corresponding paragraph). Moreover, we are planning to add more examples and analyses for the failure cases of our method.

---

> > ### Comment · Reviewer_ZsrE · 2021-08-22
> > **Thank you for the response**
> >
> > Re: Comment 2
> >
> > Is it possible that my confusion is caused by the fact that the same s_j is used for both groundtruth and prediction? If so, my concern can be addressed by updating the text using different symbols for them.
> >
> > Re: Comment 3
> >
> > My comment was to suggest to implement some of the critical previous methods into your code (probably, in many cases, only the essence of the methods that have inherent problems to be solved by the proposed method) and compare them with the proposed method rather than just running the official code by yourself. There are many factors (e.g. minor differences of hyper parameters of the optimizer) that could affect the accuracy by a couple of percent in absolute numbers, which could easily change the conclusion of the paper. If this was a competition to achieve a state-of-the-art result, this would be enough, however, we would be more interested in scientific finding for NeurIPS. I am still afraid that it is not very clear what actually contributed to the improvement by the current experimental design. I understand that this can be time-consuming and challenging. Thus, it is not mandatory, but definitely a plus.
> >
> > Re: Comment 4
> >
> > I think it would be good to denote the conditions in the paper. For example, you could consider marking all the experiments you ran by yourself with some symbol or the other way around.

---

> > > ### Author Response · Authors · 2021-08-26
> > > **Responses to the Comments of Reviewer ZsrE**
> > >
> > > Response_2:
> > >
> > > Yes. $s_j$ in Eq. (1) and (3) stands for the ground truth while $s_j$ in Eq. (4) and (5) stands for prediction. Thanks for your suggestion and we will use different symbols for them as you suggested.
> > >
> > > Response_3:
> > >
> > > The setting of hyperparameters usually depends on the method design and different methods may have different optimal hyperparameter configurations (optimizer, learning rate, code framework, etc.) respectively. It is unfair to keep hyperparameters the same or similar during comparison. Therefore, we believe that running the official codes is more meaningful than re-implementing the previous methods under the same code framework. However, to solve your concern on this point, it might be a good idea to re-implement our method under some opensource frameworks (e.g., MMOCR) which have integrated some critical methods such as DB, PAN. We are planing to do that in the future and release the source code.
> > >
> > > Response_4:
> > >
> > > Thanks for your advice. We will add the results of the experiments ran at our workstation and mark them with some symbols.

---

### Official Review · Reviewer_qKXD · 2021-07-19

**Rating:** 6
**Confidence:** 4

**Summary:**

The paper deals with the problem of scene text detection, focusing on irregular text.
The paper proposes a segmentation approach, in line with recent methods that deal with irregular text detection.
The Main contributions of the paper are:
- A detection method base on predicting both clipped segmentation masks (as other methods do) referred to as text kernels, and shifts for the areas around the kernel that can assign nearby pixels to the kernels. This allows one to perform quick postprocessing, as opposed to some other families of methods like PAN[35] or TextSnake[21].
- A modified loss to predict the shifts, that uses masking.
Empirical results on standard benchmarks show accurate results at better FPS.

**Limitations And Societal Impact:**

Supplementary material shows some limitations / failure cases. A common challenge is deciding whether two components should be connected or not.
No discussion about societal impact. Text detection has less obvious societal impact than other tasks such as face identification, but improving it and making it faster  can still have consequences,  e.g. license plate recognition in surveillance systems with lower quality hardware and non-stationary setups.

**Main Review:**

Originality / Novelty:
Paper build on top of recent methods that cast the text detection problem as one of finding segmentation proposals. As most other methods, the paper has a branch that predicts a clipped segmentation mask that is then thresholded into a binary mask, and where connected components are found to construct the "core" regions. Contrary to other methods, that then perform complex post-processing, or to methods such as Mask TextSpotter v3, that simply "undo" the clipping, this method proposes to learn, on a second branch, a shift vector on pixels around the text regions, that shifts them towards the text core. This seems to work well in practice (or at least slightly better), but it seems very incremental compared to the unclipping method.
The paper also proposes a relaxed L1 loss, but it seems to be an elementwise multiplication of the smooth L1 loss and a binary mask, similar to the loss used by many works on the segmentation branch.
Overall, the paper is incremental, and I am not sure it reaches a minimum level of originality or novelty.

Quality / clarity: The paper is, unfortunately, not very well written. The method is not well motivated, and, once challenges of current methods are explained, the paper goes straight into what are the steps of the proposed method (page 2), instead of explaining first why the proposed method would help with those problems. I had to go through the paper a couple of times before I could understand the motivation behind the described method, and how it is a smarter version of the automatic unclipping of Mask TextSpotter v3.

Significance / results: The method is evaluated on standard benchmarks and results seem to outperform state of the art methods at a good FPS. Authors mention that they will release code, which will significantly help the community. I have some questions about some specific results:

- Similar to how the proposed segmentation proposal can be fed into Mask TextSpotter v3 (Table 5, with a slight increase of accuracy), one could detach the recognition branch of Mask TextSpotter v3 and use it only for detection (e.g. Table 2 of the Mask TextSpotter v3 reports results in the MSRA-TD500 dataset).
1) Why not include the TextSpotter results in Table 4 in this paper?
2) Compared to the TextSpotter detection shown in Table 5, there seems to be a meaningful increase in accuracy when using the proposed method. However, the proposed method is more expensive since it needs to run an additional head, creating extra activations. What is the FPS and memory requirements of the TextSpotter segmentation proposals, compared to the proposed ones?
- Figure 3 mentions clustering. Is this just a plain assignment of pixels to the text core after applying the shift, or is there other operation involved?

- Nit: Figure 7a-left in supplementary material, that's actually not an error. The pattern says NONNA GELSA. The proposed method is doing better than the authors are giving it credit for.

**Time Spent Reviewing:**

3

---

> ### Author Response · Authors · 2021-08-10
> **Responses to the Comments of Reviewer qKXD**
>
> Thanks for your constructive comments. Below are our detailed responses to your concerns:
>
> Comment_1:
>
> “Originality / Novelty: Paper build on top of recent methods that cast the text detection problem as one of finding segmentation proposals. …….. This seems to work well in practice (or at least slightly better), but it seems very incremental compared to the unclipping method. ……. Overall, the paper is incremental, and I am not sure it reaches a minimum level of originality or novelty.”
>
> Response_1:
>
> Indeed, as you said, most existing arbitrary-shaped text detection methods try to predict a clipped segmentation mask, and there also exist some methods (e.g., Mask TextSpotter v3) that predict the text kernel and then expand it to the whole text instance. On the contrary, we provide a completely different manner to effectively represent a text instance with the definition that “all the pixels that can be shifted into the region of the same text kernel form a text instance”. More specifically, our method consists of two main modules: the segmentation branch and the regression branch, and we divide the text instance into two regions: the text kernel and the ignoring region. The segmentation branch aims to predict the text kernel and the regression branch tries to regress the centripetal shift of each pixel. Ideally, with the predicted centripetal shifts, all pixels in a text instance can be shifted into its corresponding text kernel. Through our delicate designs, the segmentation and regression branches can closely collaborate to precisely predict the text kernel and the centripetal shift value for each pixel. Both theoretical analyses and experimental results demonstrate that more accurate text regions can be obtained by using our proposed method compared to other state-of-the-art approaches (i.e., segmentation-centered or regression-centered approaches including Mask TextSpotter v3) since our method balances the segmentation and regression branches in a novel and more effective manner. Therefore, we believe that our paper has provided a new insight to design efficient text instance representation which plays a key role in text detection methods.
>
> Comment_2:
>
> “Quality / clarity: The paper is, unfortunately, not very well written. ……. I had to go through the paper a couple of times before I could understand the motivation behind the described method, and how it is a smarter version of the automatic unclipping of Mask TextSpotter v3.”
>
> Response_2:
>
> Thanks for your comments. We will follow your suggestions to revise the introduction section to describe the motivation of our method more clearly and intuitively. Mask TextSpotter v3 is basically a segmentation-centered approach that expands the predicted text kernel to the text instance based on the pre-defined shrink ratio. On the contrary, as mentioned above, our method can precisely predict the category of each pixel and thus obtains a more accurate text region compared to Mask TextSpotter v3, by simultaneously exploiting the advantages of both segmentation and regression branches.
>
> Comment_3:
>
> “Significance / results: ……. I have some questions about some specific results:
>
> •	Similar to …….
> 1.	Why not include ……?
> 2.	Compared to ... What is the FPS and memory requirements of the TextSpotter segmentation proposals, compared to the proposed ones?
>
> •	Figure 3 mentions ……., or is there other operation involved?
>
> •	Nit: ...…. The proposed method is doing better than the authors are giving it credit for.”
>
> Response_3:
>
> Thanks for your appreciation for our work. We promise that the source code of this paper will be released once the paper is accepted. Detailed responses to your concerns on those specific results are listed as follows:
>
> -- Indeed, as you suggested, we can “detach the recognition branch of Mask TextSpotter v3 and use it only for detection”. We will include the text detection result of Mask TextSpotter v3 in Table 4 of our revised paper. As we can see, the F-measure value of our method and Mask TextSpotter v3 evaluated on MSRA-TD500 is “86.1%” and “83.5%”, respectively, validating the superiority of our method compared to the state of the art. Yes, as you commented, when replacing the detection branch of Mask TextSpotter v3 with our model, the computational cost will be slightly increased. However, due to the large input image size and the heavy model architecture in the original configs, the increase of memory requirements and the decrease of FPS are both relatively small. We will provide detailed information regarding this issue.
>
> -- We have provided a detailed description of the inference step of our method in the third paragraph of Section 3.2, Page 6. As you mentioned, the clustering is “just a plain assignment of pixels to the text core after applying the shift” and it can be implemented by one matrix operation which is much more efficient compared to other existing approaches.
>
> -- Sorry for our mistake. Indeed, the pattern in Figure 7a-left of our supplementary material says “NONNA GELSA”. We did not notice that because the ground truth of this image in the dataset did not include this region as text instances. As you said, this example somehow actually demonstrates the effectiveness of our method for detecting some strange text patterns, instead of showing its limitation. We will correct this mistake and add more examples of our failure cases in the supplemental material.

---

> > ### Comment · Reviewer_qKXD · 2021-09-02
> > **Thanks**
> >
> > Thank you for your response. I think it addresses most of my concerns, and I've therefore upgraded my rating.

---

### Official Review · Reviewer_U3is · 2021-07-20

**Rating:** 6
**Confidence:** 2

**Summary:**

This paper presents a method for scene text detection with a neural network predicting both the probability of pixels belonging to text kernels (shrunk version of text bounding polygons) and pixels shifts towards or away from those kernels. The text kernels are predicted from the first head and extended by aggregating surrounding pixels which predicted shifts place them inside the kernel, to form the text regions. Experiments carried on various datasets show a better accuracy with faster predictions than the state of the art methods for text detection and recognition.

**Limitations And Societal Impact:**

The authors present many cases were the proposed method works well. It might be interesting to see cases where it fails along with an analysis of the possible reasons.

**Main Review:**

The paper is well written, relatively clear and easy to follow. A lot of details are given regarding the model and its training and the experiments look sound and show superior performance of the proposed solution.

However, the organisation of the paper could be improved to ease the understanding of the method. Some parts were a bit harder to follow. For example:
  - It is difficult to understand what the authors mean by centripetal shifts in the introduction. Either give less details or simplify the description
  - what is meant by "pixels that can be shifted?" in the beginning of the "Methodology" section?
  - the interest of text kernel and shifts could be made clearer earlier in the text. Maybe remind the reader why predicting text kernels is preferable to predicting the text region directly.
  - an example for the regression mask and its usefulness could be nice to better understand the motivation

Regarding the related work, we expect more details and especially how this paper can put in perspective of what already exists. Maybe include or move some parts of the introduction in the related work section. Also, it could be interesting to include more reference in the text of the introduction to support the statements made regarding existing methods.

Regarding the experiments, the results are good across various datasets and the experimental setup is well described. While this makes this simple method significant and useful to the community, we expect a bit more analysis of the method and choices made. In particular, the ablation study is not exactly an ablation study: that section compares network architectures and losses. The results are nonetheless interesting. More importantly, the choice of hyperparameters and the impact they have on the accuracy and speed, as well as the robustness of the method to variation of these hyperparameters could be really helpful to comprehend how the proposed method works and empirically justify the method design (e.g. interpolation factor in the loss, threshold for probability map, shrinking factor for the text kernels, ...)

Misc.
  - SPN is not defined in the abstract
  - introduction: slow inference speed -> slow inference?
  - introduction: fig 3 far from its first reference: maybe switch figures 2 and 3?



**Time Spent Reviewing:**

2

---

> ### Author Response · Authors · 2021-08-10
> **Responses to the Comments of Reviewer U3is**
>
> Thanks for your constructive comments. Below are our detailed responses to your concerns:
>
> Comment_1:
>
> “However, the organisation of the paper could be improved ……. For example:
> ……… -- an example for the regression mask and its usefulness could be nice to better understand the motivation”
>
> Response_1:
>
> We will follow your suggestions to further improve the organization of our paper. Specifically:
>
> -- We will follow your suggestion to modify the description in the introduction section to make readers understand the concept of “centripetal shifts” easier.
>
> -- Let us recall that we have divided a text instance into the text kernel and the ignoring region. In fact, what we mentioned in the paper is “All the pixels that can be shifted into the region of the same text kernel form a text instance”. Namely, with their centripetal shifts, pixels in a text kernel and the corresponding ignoring region can be shifted into the same text kernel. In this manner, we provide a new way to represent the text instance, and thus more efficient and effective text detection can be achieved.
>
> -- Nice suggestion! We will “remind the reader why predicting text kernels is preferable to predicting the text region directly”.
>
> -- As clearly defined in Eq. 4, the regression mask is used to mask out the correctly-predicted pixels (either text instance or background) and thus only the gradients of incorrectly-predicted pixels are not equal to 0. As you suggested, we will add a figure to show “an example for the regression mask and its usefulness” to let readers “better understand the motivation”.
>
> Comment_2:
>
> “Regarding the related work, we expect more details and ……. Also, it could be interesting to include more reference in the text of the introduction to support the statements made regarding existing methods.
>
> Response_2:
>
> Thanks for your valuable suggestions at this point! We will follow your suggestions to carefully revise the sections of related work and introduction.
>
> Comment_3:
>
> “Regarding the experiments, the results are good across various datasets and the experimental setup is well described.
> …… to comprehend how the proposed method works and empirically justify the method design (e.g. interpolation factor in the loss, threshold for probability map, shrinking factor for the text kernels, ...)”
>
> Response_3:
>
> Thanks for your suggestions. We will add more experiments to show how those hyperparameters in our method could affect the performance, including “interpolation factor in the loss, threshold for probability map, shrinking factor for the text kernels, ...”. Specifically, those new experimental results will be added into Section 4.3 by changing the original name “Ablation study” into “Effects of different model settings”.
>
> Comment_4:
>
> “Misc.
>
> •	SPN is not defined in the abstract
>
> •	introduction: slow inference speed -> slow inference?
>
> •	introduction: fig 3 far from its first reference: maybe switch figures 2 and 3?”
>
> Response_4:
>
> Thanks for your comments. We will carefully proofread the manuscript to fix all typos in our paper.
>
> Comment_5:
>
> “The authors present many cases were the proposed method works well. It might be interesting to see cases where it fails along with an analysis of the possible reasons.”
>
> Response_5:
>
> Due to the page limit, we have provided some failure cases in the supplemental material. We will add more examples and corresponding analyses in the revised version of the supplemental material.

---

### Decision · Program_Chairs · 2021-09-27

**Decision:**

Accept (Poster)

**Comment:**

This paper presents a new method for scene text detection and recognition based on the integration of individual local responses and their centripedal shifts.

The paper has received 4 expert reviews, which were quite positive, with in particular one reviewer championing the paper. While the reviewers (and the AC) agreed, that the paper had weaknesses in presentation and writing, there was a general agreement that the paper has merits, sufficient novelty, and convincing results.

The AC concurs and proposes acceptance.